# Expanded Electroluminescence in High Load CdS Nanocrystals PVK-Based LEDs

**DOI:** 10.3390/nano9091212

**Published:** 2019-08-28

**Authors:** Fernando Rodríguez-Mas, Juan Carlos Ferrer, José Luis Alonso, Susana Fernández de Ávila

**Affiliations:** Communications Engineering Department, Universidad Miguel Hernández, 03202 Elche, Spain

**Keywords:** cadmium sulfide, PVK, hybrid light-emitting device, electroluminescence, nanocrystals, solution process, QDs-LED

## Abstract

Immiscibility between dimethyl sulfoxide (DMSO) and polar solvents used for poly(*N*-vinylcarbazole) (PVK) solutions, leads to failed light-emitting diodes when colloidal cadmium sulfide (CdS) nanoparticles capped with thiophenol are incorporated to their active layer. To prevent this, a heat treatment is applied to the CdS nanoparticles in order to evaporate DMSO solvent. After evaporation most of the nanoparticles increased their size, and some of them show hexagonal crystalline structure instead of the original cubic zinc-blende observed in colloidal pre-treated nanoparticles. Nevertheless, enhanced electronic properties are measured in light-emitting devices when DMSO-free nanoparticles are embedded in the poly(*N*-vinylcarbazole) active layer. Light emission from these hybrid devices comprises the whole visible range of wavelengths as searched for white LEDs. Moreover, electroluminescence from both types of CdS nanoparticles (smaller cubic and bigger hexagonal) has been discriminated and interpreted through Gaussian deconvolution.

## 1. Introduction

During the last decade, organic light-emitting diodes (OLEDs) have been extensively studied by the international scientific community [1,2,3,4]. This kind of device has attracted great interest due to their adjustable optical and electrical properties [5,6], mechanical flexibility [7,8], low cost [8], and simple manufacturing processes [9].

A research area within the OLEDs is the inclusion of semiconductor nanoparticles [10,11]. The blend of organic polymers with inorganic semiconductor nanocrystals (NCs) presents potential advantages and the chance to design a device with the benefits of both, organic and inorganic materials [12,13,14].

Several deposition techniques, such as chemical vapor deposition (CVD) [15], ink-jet printing [16] or spin-coating [17], may be employed to grow the OLED devices, which have a basic structure consisting, from bottom to top, of transparent substrate and electrode, a hole injection layer, a polymer active layer and a cathode. In this paper, poly(3,4-ethylenedioxythiphene):poly (styrenesulfonate) (PEDOT:PSS) has been used as hole injection layer, and a hybrid organic-inorganic nanocomposite, composed by a blend of cadmium sulfide (CdS) nanoparticles in a matrix of poly(*N*-vinylcarbazole) (PVK), as active layer.

The inclusion of nanoparticles has been demonstrated to improve the devices properties. A thin film of lead selenide (PbSe NCs) in a poly[(3-methoxy,5-octoxy)-1,4-phenylenevinylene] (MOOPV)-based device increased the current density from 0.068 to 0.309 mA/cm^2^ [18]. The structure of indium tin oxide (ITO)/PbS NCs/[6,6]-phenyl-C61-butyric acid methyl ester (PCBM)/Ag presented an enhancement in the fill factor due to the nanocrystals [19]. In particular, the CdS NCs are often used, but improvements are needed. In poly(3-octylthiophene) (POT)-based solar cells, the CdS NCs improved the power conversion efficiency (PCE) from 0.086% to 0.581% [20] and, in poly(3-hexylthiophene) (P3HT):PCBM-based devices, nanocrystals enhanced the PCE from 2.95% to 4.41% [21]. Concerning LEDs, CdS nanoparticles can be used to modify the CIE 1931 coordinates of the light emission although it is usual to find reduced brightness of the emitted light and poor efficiency in simple structured hybrid LEDs [12]. Nevertheless, it is possible to improve efficiency of hybrid LEDs using a band-gap engineering method and multilayered structures [22]. Zinc oxide nanoparticles (ZnO NCs) in a PVK matrix have lead to devices with improved charge transport because ZnO NCs helped the electron injection [23].

Hybrid LEDs with CdS NCs deposited as a separated layer on a PVK film have been reported showing light emission where the luminescence of PVK is absorbed and mitigated by the emission of the nanoparticles. Near-white LEDs are manufactured with this technique [12,24]. In contrast, hybrid PVK based LEDs with CdS NCs in the same layer will produce electroluminescence where emission of both PVK and nanoparticles are visible. In this way, hybrid LEDs with CIE coordinates closer to white could be manufactured.

In this paper we use a very basic layer structure avoiding the use of hole or electron injection layers or electron transport layers so the efficiency expected in these LEDs is low. Current efficiencies found in the literature for simple structure devices with blended hybrid active layer have values of 0.078 cd/A for a PVK-CdSe NCs LED [25] or 0.002 cd/A for PVK-near-IR dye-CdSe/CdS NCs LED [26].

We have used an easy technique for the synthesis of CdS NCs, via thiolate decomposition [27,28]. Thiophenol has been used as ligand because their aromatic rings enhance the transport of the excitons [29] towards surrounding polymers and NCs.

Due to the excellent solubility of thiophenol in dimethyl sulfoxide (DMSO), colloidal CdS nanoparticles synthesized by thiolate decomposition end up dissolved in DMSO. Having the nanoparticles dissolved in DMSO generates some inconveniences, because PVK cannot be dissolved in DMSO and, besides, this polymer is typically processed in solvents that are not miscible with DMSO. This fact leads to an inconvenience since two immiscible solvents must be blended to prepare the solution for casting the active layer doped with nanoparticles. This mixture causes homogeneity problems in the spin-coating process since DMSO is a solvent with higher viscosity and evaporation temperature than toluene. Most of the toluene is evaporated in the rotation process inherent to the spin-coating, but instead, DMSO remains in liquid state much longer dragging the nanoparticles to that phase. As a consequence, poor morphology of the films is observed and the fabrication of LEDs based on this process failed, as we show in Section 3.3. To achieve successful devices, we present in this paper a simple procedure to prepare free-DMSO CdS NCs through evaporation, helping to homogenize the blend for hybrid active layer (PVK:CdS NCs), as a necessary condition for good quality LEDs prepared by spin coating. After this evaporation treatment, the synthesized nanoparticles are obtained as a dry powder helping the control of the CdS amount incorporated to the devices.

We have studied the alterations observed in the CdS NCs after evaporation and fabricated a set of hybrid PVK-based LEDs with these CdS NCs embedded in the active layer. We have studied the electronic and electroluminescent properties of such devices, analyzing the influence of the quantity of nanoparticles in the devices, and correlating the results with the CdS NCs structural characteristics.

## 2. Materials and Methods

### 2.1. Materials

Cadmium nitrate-tetrahydrate (Cd(NO_3_)_2_·4H_2_O, 99.99%), thiophenol (99%), sulfur powder (99.98%), poly(9-vinylcarbazole) (PVK, 98%), poly(3,4-ethylenedi-oxythiophene):poly(styrene sulfonate) (PEDOT:PSS, 1.3% water solution), toluene, methanol and dimethyl sulfoxide (DMSO) were purchased from Sigma-Aldrich (Darmstadt, Germany) and used without further purification.

### 2.2. Synthesis of Colloidal CdS Nanoparticles in DMSO

CdS nanoparticles have been synthesized according to the thiolate decomposition method [18,19]. Cadmium thiophenolate, Cd(C_6_H_5_–S)_2_ was dissolved in DMSO. The *m*_[Cd(C6H5–S)2]_/*V*_[DMSO]_ = 40 mg/mL relationship is used. In another flask, 0.175 g of sulfur powder was dissolved in 20 mL of toluene, 1% in weight. When the first solution was homogenized, S (1%) was added. We prepared a first synthesis solution keeping a ratio of 5 mL of sulfur solution S (1%) for each gram of cadmium thiophenolate: *V*_[S(1%)]_/*m*_[Cd(C6H5-S)2]_ = 5 mL/g. A couple of minutes after mixing, the solution changed to transparent blend indicating that CdS(col) nanoparticles had been synthesized in DMSO.

### 2.3. Hybrid LEDs Fabrication

Hybrid light-emitting diodes with a layer sequence of ITO/PEDOT:PSS/PVK:CdS/Al were fabricated by spin-coating. Commercial glass substrates covered with a semitransparent ITO layer, were routinely cleaned by sequential ultrasonication in 1,2,4-trichlorobenzene, acetone and isopropyl alcohol, and then dried with N_2_.

Aqueous PEDOT-PSS dispersion was spin-coated (6000 rpm) on the clean ITO surface and then annealed at 100 °C for 60 min. Then, the active layers were spin-coated at 4000 rpm and dried at 80 °C for 60 min.

Finally, the metallization of the cathodes was performed evaporating aluminum in a high vacuum chamber (10^−6^ mbar) until 200 nm of thickness were achieved.

### 2.4. Characterization

The measurements of optical absorption were implemented with a T92+ UV/VIS spectrophotometer from PG instruments Ltd. (Lutterworth, United Kingdom) and the measurements of photoluminescence (PL) were performed with a Modular Spectrofluorometer Fluorolog-3 from Horiba Scientific (Madrid, Spain). In all the photoluminescence measurements, the excitation wavelength was fixed at λ_exc_ = 365 nm.

Atomic Force Microscopy (AFM) was performed using a NTEGRA Probe NanoLaboratory (Limerick, Ireland) working in tapping mode.

Thermal Gravimetric Analysis (TGA) and Differential Scanning Calorimetry (DSC) were done using a Mettler-Toledo TGA/SDTA851e/SF/1100 (Barcelona, Spain).

Transmission electron microscopy (TEM) analysis was performed by Jeol 2010 (Tokyo, Japan) operating at 200 kV.

Current density vs. voltage (J–V) curves of the LEDs were measured using a Keithley 2400 Sourcemeter equipment (Bracknell, UK).

Electroluminescence (EL) characterization was performed with a Triax 190 monochromator (Madrid, Spain) and a multichannel thermoelectrically cooled CCD Symphony detector by Horiba Jobin Yvon (Madrid, Spain).

## 3. Results

### 3.1. Nanocomposites PVK-Colloidal CdS NCs

Our first goal was to incorporate the synthesized CdS NCs as part of the active layer in hybrid LEDs based in PVK polymer. To this purpose we prepared the active layer nanocomposite mixing the solution of colloidal CdS NCs in DMSO with a solution of PVK in toluene. The solution for LEDs active layers had a PVK:CdS mass ratio of 1:1 with a concentration of 3% by weight of the hybrid solution. In addition, a reference of PVK in toluene, without nanoparticles, was also prepared at 3 wt%. Optical absorption and PL measurements were carried out in these solutions.

Figure 1 shows the absorption curves corresponding to pristine PVK and hybrid PVK:CdS(col) solutions. We observed that the solution with nanoparticles exhibit an evident increase in absorbance with respect to the PVK sample for wavelengths above 370 nm. The background absorption at visible wavelengths increases for the hybrid solution in the whole range. The absorption edge due to the presence of CdS NCs can be seen as a shoulder appearing around 400 nm. The inset in Figure 1 shows absorbance and PL spectra of a solution of colloidal CdS NCs in DMSO. A value of 3.253 eV (381.17 nm) was obtained for the absorption band edge. The maximum for light emission, with an excitation wavelength of 365 nm, is found at 480 nm.

For CdS NCs synthesized at low temperatures, it is common to observe broad photoluminescence emission bands attributed to trap emission from deep surface states primarily resulting from defects in the nanocrystals surface. The most frequent defects in CdS NCs are sulphur or cadmium vacancies, interstitial sulphur and cadmium atoms adsorbed on the NP surface [30]. The big distance between emission wavelength and absorption onset, close to 100 nm in the inset of Figure 1, is indicative of such emission from deep surface states in the NCs.

The size of the CdS nanoparticles can be estimated by the equation suggested by Brus [31]:
*E_n_* = *E_b_* + (*ħ*^2^π^2^/2*R*^2^) × (1/*m_e_**+1/*m_h_**) − 1.8e^2^/(4πε_0_ε*R*),
(1)
where *E_n_* is the energy gap of CdS nanoparticles, and *E_b_* = 2.42 eV is the energy gap of bulk material, ε = 5.7 is the dielectric constant, *m*_e_* = 0.19*m*_0_ is effective mass of electron, and *m_h_** = 0.80*m*_0_ is effective mass of hole (*m*_0_ is the rest mass of electron).

The estimated diameter of these colloidal CdS NCs is shown in Table 1.

The normalized PL spectra of the pristine PVK and hybrid solutions are plotted in Figure 2. The excitation wavelength is λ_exc_ = 365 nm. The hybrid curve (red) presents a main peak related to PVK emission and the influence of the CdS NCs can be perceived as a shoulder at longer wavelengths. The solution without CdS shows the PVK characteristic emission peak at 400 nm corresponding to the excimer emission of PVK [13]. In the solution with nanoparticles, the PL is defined by a strong emission peak at approximately 415 nm related to the polymer emission. The red-shift in emission wavelength might be due to well-known dependence on solvent polarizability. DMSO is a highly polar solvent, so the distance between polymer molecules decreases, therefore increasing local polarization field and producing the shift observed with PVK emission [32].

The asymmetry of the peak on the longer wavelength side is generated by the influence of the CdS nanoparticles. The emission intensity coming from the NCs is low and it is expected at wavelengths very close to the emission from the PVK, thus its effect is hidden for this ratio of NCs.

### 3.2. Hybrid LED with CdS Colloidal Nanoparticles

Hybrid light-emitting diodes with the configuration shown in Figure 3 were fabricated following the procedure described in Section 2.3. The previously characterized solutions were used as the starting material for the emissive layers. CdS NCs in DMSO were mixed with PVK dissolved in toluene with 1:1 mass ratio. The concentration of this solution was at 3 wt%.

Two types of LEDs were manufactured. The first one is the reference, organic LED with pristine PVK emissive layer. In the other one, the active layer consists of the mixture of PVK and CdS(col) NCs.

We measured the current density vs. voltage (J–V) curves of these LEDs. The results are shown in Figure 4A. Hybrid LEDs using CdS NCs in DMSO solution presented a resistive behaviour and they did not show any electroluminescence. Except for the pristine PVK LED, all the hybrid devices we fabricated using colloidal CdS NCs did not exhibit a uniform surface.

Given the results of the J–V curves and the absence of light emission for these devices, characterization by atomic force microscope (AFM) was performed to observe in detail the surface of the hybrid devices (see Figure 4B).

In the spin-coating technique, the solvent is removed from the film due to the speed of rotation as well as the time used in the process. However, the density and viscosity of solvents are factors to take into account. In order to not modify the conditions of the manufacturing route, the rotation speed, and the time of spinning were exactly the same for all the fabricated devices.

In the spinning process, the solvents undergo the gravity and the centrifugal force, and a viscous force caused by the solvent viscosity. This viscous force is a friction force, contrary to the centrifugal force. The density of the solvent increases the gravity and centrifugal force and the viscosity increases the viscous force. The density/viscosity ratio conditions the spin-coating process. For toluene, this ratio is 1.47 whilst for DMSO it is only 0.55. In films where both solvents are present, the difference in these ratios produces artifacts in the active layer. The solvent with the highest density/viscosity ratio tends to remain in the lower part of the layer and/or it is quickly spelt out of the substrate. DMSO could prevent a proper elimination of toluene because its density/viscosity ratio is lower and disturbs toluene mobility, causing solvent areas of toluene that are not eliminated, remaining in the film under DMSO. On the other hand, DMSO with a lower density/viscosity ratio is prone to remain in the upper part of the liquid film and last longer over the film surface prior to be expelled from the substrate. Solvent areas could be eliminated by increasing the spin speed or time in the process, but increasing these parameters, the nanoparticles are probably blown away.

With subsequent heating, the solvent residues evaporate. The toluene areas are evaporated and very small bubbles remain in the active layer. When the toluene evaporates, the tiny craters (black spots) that we observe in Figure 4B are created. A statistical study has been performed and the craters have an average depth of 12.5 ± 0.6 nm. During the evaporation of the cathode, aluminium is introduced into the craters, causing narrowing between anode and cathode and micro short circuits. This situation can create the resistive behaviour that we have observed in the J–V curves, see Figure 4A, because electrons flow may have a preference for narrowing and not for the thicker active layer. This can also be a reason why these devices do not show electroluminescence.

Furthermore, in Figure 4B, some isolated circular monticules, bright white spots of hundreds of nanometres diameter, are observed. These white protuberances, close to 100 nm height, can be agglomerations of nanoparticles because the nanoparticles surrounded with thiophenol ligands are soluble only in DMSO and they tend to remain in this solvent. If microscopic drops of DMSO hold on the surface of the sample after the spinning process, the solvent will evaporate by post-annealing and the NCs in DMSO will agglomerate forming these peaks on the top of the layer.

The confluence of high peaks of, presumably, CdS NCs, and tiny craters in the PVK matrix within a short distance, is probably the cause to these hybrid LEDs malfunction.

### 3.3. Preparation of Evaporated CdS Nanoparticles Powder

The inadequate morphology of the active layer, found in the devices manufactured in the previous section, is possibly due to the presence of two solvents with quite different properties. To eliminate the DMSO present in the active layer solution, we have tested a very simple method to obtain nanoparticles to embed in the PVK matrix. Instead of performing a complex, time consuming and more expensive ligand-exchange process to remove thiols from the NCs surface and replace them with other ligand soluble in toluene, we decided to simply evaporate the DMSO from colloidal CdS NCs solutions by heating.

Thermal gravimetric analysis (TGA) and differential scanning calorimetry (DSC) were performed to check the cadmium thiolate stability with the temperature. The results are displayed in Figure 5. To eliminate the solvent, a temperature of 200 °C was chosen, because this temperature does not produce changes of mass or state in Cd(C_6_H_5_S)_2_ (Figure 5) and is higher than DMSO boiling point (189 °C). For this experiment we used solutions of CdS NCs synthesized in DMSO, like the one studied in Section 2.2. CdS(col). One day after the synthesis process, the solution was heated at 200 °C on a heating plate. Once the solvent was evaporated, a yellow powder of CdS NCs is gathered in the bottom of the vessel. Dry evaporated CdS nanocrystals with their capping are obtained CdS(eva).

Powder of CdS (eva) NCs was redispersed in toluene and optical absorption and photoluminescence measurements were carried out and compared to CdS(col) measurements. According to Youssef et al. [33], the thermogravimetry of CdS nanoparticles shows that significant changes due to temperature occur at temperatures quite higher than 200 °C, mostly around 275 °C. Below that temperature it should be expected that heating does not modify the structure of the nanoparticles.

Nevertheless, in our experiment, the heating received during the evaporation causes the photoluminescence to be activated increasing photoluminescence intensity. In addition, a red shift of the maximum position of the CdS emission peak is observed.

Figure 6 shows the comparison between normalized PL emission of colloidal CdS NCs as synthesized in DMSO (black squares) and these dried CdS(eva) NCs redispersed in toluene (blue squares). The CdS emission peak shift is clearly seen. The colloidal CdS nanoparticles synthesized directly in DMSO have their maximum emission at 480 nm while for evaporated CdS NCs redispersed in toluene, the maximum is located at 586 nm. Moreover, a PL emission enhancement is clearly observed. This increased PL efficiency after heating has been previously reported in CdS NCs [34] and attributed to reduced local strain and nonradiative decay centres after annealing. Optical absorption was also registered for these evaporated NCs dispersed in toluene, and compared, in Figure 7, with the absorption of the NCs prior to evaporation.

Figure 7 corroborates that evaporation produces also a shift of absorption edge towards a longer wavelength and a lower energy gap. Comparative values of absorption edge for both types of NCs are collected in Table 1. The increase of absorbance observed for evaporated NCs, especially at longer wavelengths, is due to the fact that these NCs are dispersed in toluene, where they are not soluble, producing high scattering.

An estimation of the evaporated CdS NCs size is done using Brus Equation (1), and the results also included in Table 1. According to this, the evaporation of the DMSO solvent goes along with the increase in size of the NCs.

In order to check the different size of NCs before and after the evaporative heating, transmission electron microscopy (TEM) analysis has been performed. To perform TEM analysis, a drop of the NCs solution was deposited on a carbon grid and then dried at room temperature.

For the as synthesized CdS(col) NCs in DMSO a TEM study with more than 90 nanoparticles was conducted. An average size is calculated for these nanoparticles of 3.07 ± 0.07 nm (sample in Figure 8A). In Figure 8B, the same study was performed for the DMSO-free cadmium sulfide NCs CdS(eva). In this case, the obtained average size is 4.12 ± 0.12 nm. A wider size distribution is found in the samples with evaporated nanocrystals. Figure 8C shows the histogram correlating DMSO-free CdS(eva) NCs with their size measured by TEM. Besides a concentration of NCs with sizes around 3 nm of diameter we observed an appreciable number of CdS NCs with diameters above 4 nm in diameter.

The heating of the nanoparticles produces a growth of their diameter from 2.9 to 4.9 nm, according to Brus equation (Table 1). This growth has been empirically demonstrated by the TEM analysis (Figure 8). The increased size also causes a shift of the PL emission and the absorption edge to higher wavelengths.

Although temperature below 200 °C has been used to carry out the evaporation of DMSO solvent and the drying of the NCs to obtain them in powder form, it is possible that several thiol chains have decomposed during the heating process. As a hypothesis, the loss of some thiol ligands around the NP surface would allow the coalescence of some of the original nanocrystals, producing the increased size observed from the evaporated CdS NCs. Notwithstanding, evaporated CdS NCs are again perfectly soluble in DMSO, thus they keep their thiol capping layer after the treatment.

In addition to the size, we observed structural changes by measuring the interplanar distance of several NCs seen in TEM images. In Figure 8A, NCs present cubic blend-type crystalline structure and {200} planes with 2.9 Å separation are highlighted. In Figure 8B, corresponding to CdS NCs after evaporation process, a wider spread of sizes is registered (Figure 8C) and crystalline planes corresponding to hexagonal besides the cubic structure are measured. Planes {$10\bar{1}0$} of hexagonal crystalline NCs are identified in Figure 8B with 3.58 Å lattice spacing.

Structural transformation of CdS NCs has been previously referenced linked to size variations. It is possible that the heat received during the evaporation gave enough energy to NCs to join between them and grow, producing stacking faults which might transform the initial and metastable cubic zinc blend structure into the more stable hexagonal wurzite-type as it has been previously reported [35,36].

### 3.4. Nanocomposites PVK-Evaporated CdS NCs

In order to test the behavior of the evaporated CdS NCs once embedded in PVK, we prepared and characterized several hybrid solutions redispersing the NCs studied in Section 3.3 in a solution of PVK in toluene.

Four solutions were prepared as indicated in Table 2. Evaporated CdS NCs were mixed with the PVK with different proportions. Then the mixed solutes were dissolved in toluene providing a fixed concentration of 3 wt%.

Figure 9 shows the absorption spectra of pristine PVK and PVK with evaporated CdS NCs solutions with different CdS contents. Absorption of the solution increases with the incorporation of CdS(eva) NCs. The ratio of CdS nanoparticles to PVK weight influences this increase. When CdS proportion in solutions increases, they evolve towards turbid solutions. This originates the raised optical absorption at the full range of wavelengths.

In Figure 9, we observe two successive shoulders: one around 380 nm and the other around 450 nm wavelength, in contrast to Figure 1 where only one step shoulder was observed at an approximately 380 nm wavelength. This observation supports the idea of CdS(eva) NCs with two different crystalline structures mixed in these hybrid solutions [35].

Figure 10 shows the PL spectra of PVK and the hybrid solutions. The habitual PVK photoluminescence quenching with increasing CdS NCs content is visible in Figure 10A. Besides the natural decrease in emission intensity due to the lowering content of the PVK polymer in the solutions, the quenching is attributed to charge transfer between the polymer and the CdS nanocrystals [14]. When photons are absorbed, electrons in PVK are excited into the lowest unoccupied molecular orbital (LUMO) of the polymer. When CdS NCs are added to the solution, these excited electrons in PVK can choose between returning to the polymer highest occupied molecular orbital (HOMO) through a radiative decay, or migrate from PVK to the conduction band of CdS NCs. Such possibility of charge transfer reduces the PVK luminescence [37].

In the inset of Figure 10, we can see the photoluminescence of the CdS(eva) NCs normalized to the maximum of PVK emission. We observe that, even though the intensity of emission coming from CdS NCs is not directly proportional to their content in the solutions, the presence of these NCs in the solutions is responsible for the luminescence increasing around 575 nm wavelength. Similar results have been reported with increasing loading of CdSe/ZnS NCs in PVK matrix [38]. The shape and position of this broad peak, almost identical to the one observed for the evaporated CdS NCs without PVK, indicates that it is again emitted from deep traps located in the surface of the nanocrystals.

### 3.5. Hybrid LEDs Fabrication with Evaporated CdS Nanoparticles

Four types of LEDs were manufactured following the same procedure described in Section 3.3, with the layered structure shown in Figure 4. One control device was fabricated with a pristine PVK emissive layer, while the active layer for the other LEDs was spin casted from the hybrid solutions characterized in Section 3.5. This time, evaporated CdS nanocrystals were used to avoid the presence of DMSO. Toluene was the only solvent used for the emissive layer in all the LEDs of this experiment.

After device fabrication, we measured the current density—voltage (J–V) curves. In addition, electroluminescence (EL) characterization was performed.

In the J–V curves (Figure 11A), it is observed that at the same voltage values, the current density increases with the inclusion of nanoparticles. The improvement is proportional to the nanoparticles increase. The evaporated CdS NCs reduce the threshold voltage (Figure 12) that is the lowest for the device with PVK:CdS ratio [1:2] and it is the highest for the LED without CdS NCs. At the same time, Figure 12 shows the current density measured at the threshold voltage. The electronic transport in these devices improves with the presence of nanoparticles.

In devices with pristine PVK active layer, the potential barrier for electrons injection is 2 eV. As represented in Figure 13, the PVK lowest unoccupied molecular orbital (LUMO) is located at −2.3 eV and for aluminium is −4.3 eV [39]. Embedding CdS NCs in the polymer matrix decreases the energy barrier for electron injection. On the basis of the effective mass approximation the electron affinity of CdS NCs depends mostly on the energy of the conduction band edge. The band gap for the evaporated CdS NCs, calculated from optical absorption, is 2.65 eV (Table 1). If the valence band edge of CdS is set on −6.8 eV relative to the vacuum level, then the energy level of the conduction band would shift up to −4.15 eV, much closer to the aluminium work function than the PVK LUMO. This approach causes the improvement of electrical conduction. Besides, electrons move through CdS conduction band with enhanced electron transport properties than those through PVK. The extension of the areas with potential closer to aluminium, increases with the ratio of CdS nanoparticles in the active layer, improving the electrical conduction with the nanocrystals content. In the hybrid LEDs of Section 3.3, with colloidal CdS nanoparticles in DMSO solution their conduction band would be located at −3.55 eV and the potential barrier would not be so close to aluminium as with evaporated CdS. Therefore, the electronic transport of the hybrid LED with evaporated nanoparticles presents electrical improvements compared to LEDs with colloidal CdS NCs. In addition, although the TGA analysis indicates that nanoparticles do not suffer alterations due to the temperature, previous TEM and optical characterization analysis indicated that several thiols may have been released from the nanoparticles. This absence of ligands favours a direct connection between PVK and the CdS core, improving the electrical behaviour.

AFM analysis was performed also for these devices with free-DMSO CdS NCs. A portion of the surface of the LED with PVK:CdS [1:2] is shown in Figure 11B. The tiny craters and circular monticules present in the devices with hybrid PVK:CdS(col) NCs layers (Figure 4B) are not observed in LEDs with evaporated CdS nanoparticles. Extensive agglomerations of nanoparticles are visible on the surface of the device instead, especially in this device that had the higher NCs filling factor (theoretically 66.6% of the layer weight). Although the roughness of the surface is important these devices presented electroluminescent emission as it is shown in Figure 14.

The efficiency of pristine PVK LED was 0.69 cd/A while for hybrid-LEDs doped with CdS NCs were 0.13 cd/A for [1:0.5], 0.09 cd/A for [1:1] and 0.06 cd/A for PVK:CdS(eva) [1:2]. The efficiencies showed that the nanocrystals reduce the external current conversion, as anticipated before [12,25,26]. Agglomeration of NCs, as observed in AFM images (Figure 11B), leads to inefficient energy transfer between the PVK matrix and thiophenol capped CdS NCs reducing luminescence efficiency [38].

Electroluminescence measurements were carried out to PVK reference and to hybrid PVK:CdS LEDs (Figure 14) in the same conditions used for LEDs in Section 3.3. The inclusion of nanoparticles in the active layer caused a decrease in the integrated emission intensity of electroluminescence, as observed in the inset of Figure 14.

For the PVK reference device, a narrow distinct peak at 412 nm wavelength is easily observed. As explained by Ye et al. [40], PVK is a strongly polar polymer. The presence of an electrical field in electroluminescence characterization enhances the predominance of the fully overlapping conformation of carbazole groups (f-PVK), where both carbazole groups are aligned face-to-face in adjacent positions along the polymer backbone. This emission energy is slightly reduced, in comparison to that from photoluminescence, due to the shortened intermolecular distance in f-PVK conformation. This explains the red shift compared with former PVK photoluminescence emission.

Additionally, next to the main emission peak, two shoulders are perceived at longer wavelengths in the EL curve of the PVK LED (black curve in Figure 14).

In order to clearly obtain the position of the peaks contributing to each EL curve, Gaussian deconvolution was performed for each spectrum (see Figure 15). Three differentiated peaks were allowed to describe each EL curve. In Table 3, the central wavelengths of the peaks obtained from the Gaussian deconvolutions have been collected.

The higher shoulder registered for the reference PVK device comes from the phosphorescence emission of the polymer [40]. For EL processes the proportion of triplet states is increased compared to PL processes, this allows the observation of phosphorescent emissions at room temperature. For the device shown in Figure 15A this emission was located at 477 nm according to our Gaussian simulation.

Finally, the lower shoulder observed in PVK EL spectrum is attributed to the electromer of PVK [40]. This emission is only visible in EL, not in PL, because photoexcitation does not normally generate the free charge carriers needed to form this complex.

Regarding the electroluminescence from the hybrid LEDs with CdS evaporated NCs, we also observed three contributions, which we have correlated by means of the Gaussian deconvolution (see Figure 15B–D).

The peak located at shorter wavelengths is coming from the PVK matrix emission. Its contribution to the whole electroluminescence goes down when the ratio of CdS NCs is increased. There are two main reasons for this expected behaviour: one just related to the lower amount of polymer present in the emissive layer when the mass of NCs increases, and the other is related to the quenching of PVK luminescence [41]. As can be observed in Figure 13 the energy difference between the conduction band of CdS NCs and the LUMO of PVK host is close to 2 eV hindering the flow of electrons towards the polymer. Thus the electroluminescence quenching might also be ascribed to the electron-trap role of the CdS NCs.

The highest peak and the shoulder observed at longer wavelengths in hybrid EL spectra are due to CdS NCs emission as can be guessed looking at Figure 14, where the increase of these contributions is noticeable along with the ratio of CdS NCs embedded in the devices.

As we have previously observed in TEM evaluation, after evaporation, some CdS NCs have grown up and changed their crystalline structure evolving from cubic to hexagonal structure. Through Gaussian deconvolution we are able to link the stronger emission from bigger CdS NCs with the longer wavelength shoulder (close to 600 nm) of EL spectra, and the weaker contribution of cubic CdS NCs, adding to raise the EL emission to its maximum intensity at wavelengths around 550 nm for the hybrid LEDs.

Thus the second peak obtained from the Gaussian deconvolution for the hybrid LEDs, is attributed to the CdS NCs that have suffered a minimum alteration after the drying process. It corresponds to the smaller CdS nanocrystals that hold on the cubic structure. This emission is assumed in the range between 524 and 552 nm (Table 3) for the LEDs shown in Figure 14 and Figure 15. There is some red-shift compared to the photoluminescence emission of the CdS NCs synthesized in DMSO (Figure 1 and Figure 2) but this is expected for electroluminescence, which is produced by different mechanisms than photo induced luminescence [42].

Finally, we assign the Gaussian curve responsible of the shoulder observed at longer wavelengths in the EL spectra of hybrid LEDs to emission from the larger size evaporated CdS NCs with hexagonal crystalline structure. This is the main contribution to the electroluminescence of these devices. The electroluminescence of these larger NCs is also slightly red-shifted if we contrast with the photoluminescence emission registered around 590 nm in Figure 6 and Figure 11. The wide Full Width at Half Maximum (FWHM) might be caused by the high size dispersion that we have demonstrated by TEM study.

In order to further demonstrate this hypothesis instead of the well-known identification of the middle peak (located around 540 nm) as the CdS NCs band-gap emission, and the longer wavelength band (located around 600 nm) as the emission related to traps located in the NC surface, we analysed the electroluminescence as a function of voltage (see Appendix A). We have checked that both peaks related to NCs follow similar trends with increased voltage. Moreover, we do not observe saturation on the emission of the longer wavelength peak at higher voltages. On the contrary, it increases slightly its influence on the emission at higher applied voltages. Based on all these evidences, we corroborate the validity of our argument against the expected band-gap and related traps emission.

In addition, we studied the CIE 1931 chromaticity for the hybrid-LEDs fabricated (see Appendix A). The color coordinates for PVK-LED were (0.24, 0.22). Increasing loading of CdS NCs modified the CIE coordinates of PVK LED towards near-white color CIE coordinates (0.33, 0.33) [21]. The chromatic coordinates were: (0.32, 0.34) for PVK:CdS [1:0.5]; (0.34, 0.38) for PVK:CdS [1:1] and (0.38, 0.39) for PVK:CdS [1:2]. Increments of CdS reduced the influence of PVK, as a result, the CIE coordinates were shifted by the presence of CdS NCs towards the coordinates of CdS NCs light emission (0.43, 0.45).

## 4. Conclusions

We have synthesized colloidal CdS nanoparticles (CdS(col)) by decomposition of thiolates in order to be introduced in the PVK active layer of OLED structures. The devices manufactured with the active layer formed by PVK and CdS (col) mixture showed poor results which are related to the immiscibility of the nanoparticles solvent (DMSO) and the polymer solvent (toluene). To overcome the shortcomings of the use of two immiscible solvents a thermal treatment was applied to the colloidal nanoparticles CdS(col), previously to their mixing with PVK, where the DMSO solvent is evaporated.

Some dried CdS NCs (CdS(eva)) showed size increases and a structural change was observed via TEM—some of them evolving from cubic to hexagonal crystalline structure, confirming a behavior previously reported for similar CdS NCs.

We fabricated hybrid LEDs with the active layer formed by PVK and CdS(eva) nanoparticles. As demonstrated in this paper, the inclusion of DMSO-free CdS NCs in the active layer improves the device’s electrical properties. Particularly, it improves the J–V curves and reduces internal resistance and threshold voltage. In addition, an expanded electroluminescence emission spanning from approximately 400 to 800 nm is obtained in these hybrid devices. Light emission related to CdS nanocrystals showed two separated sources. Gaussian deconvolution analysis along with TEM and optical characterization allow us to attribute each contribution to one of the two types of CdS nanocrystals: cubic zinc-blende and hexagonal wurzite structure, respectively.

As far as we know, this is the first time that electroluminescence coming from cubic as well as hexagonal CdS nanocrystals has been noticed and identified in a single emissive layer LED.

The activation of the electroluminescence from the DMSO-free CdS NCs opens the possibility to fabricate optimized white LEDs with nearly uniform emission intensity in the whole visible range of wavelengths. In tailoring the thickness of the active layer, PVK:CdS NCs ratio, and improving the multilayer structure by means of band-gap engineering, current efficiency values should be readily increased.

## Figures and Tables

**Figure 1 nanomaterials-09-01212-f001:**
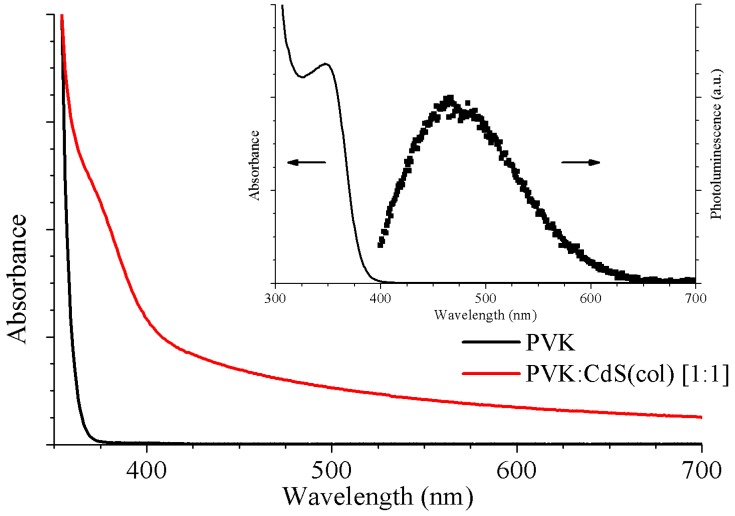
The absorption spectra of the (black) PVK and (red) hybrid PVK:CdS(col) solutions. The inset shows UV-Vis absorbance and photoluminescence from CdS(col) NCs in DMSO.

**Figure 2 nanomaterials-09-01212-f002:**
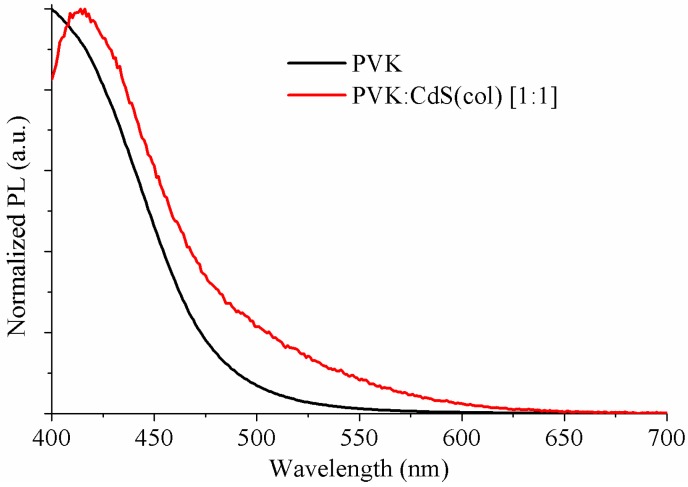
The normalized PL spectra of the active layer solutions: (black) PVK and (red) PVK:CdS(col) [1:1].

**Figure 3 nanomaterials-09-01212-f003:**
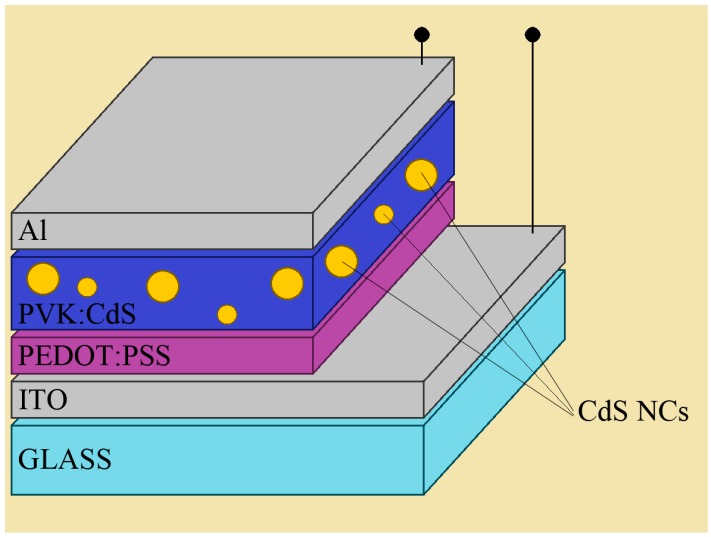
Schematic structure of hybrid LED with PVK:CdS active layer.

**Figure 4 nanomaterials-09-01212-f004:**
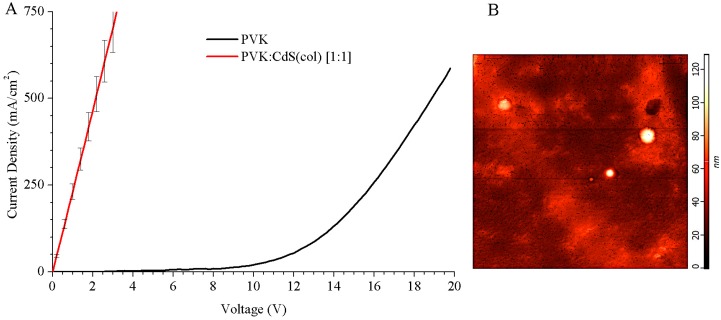
J–V curves of the OLEDs (**A**) and atomic force microscopy (AFM) image (area of 10 µm × 10 µm) (**B**) corresponding to the hybrid LED with PVK:CdS(col) active layer. The color scale on the right indicates surface roughness.

**Figure 5 nanomaterials-09-01212-f005:**
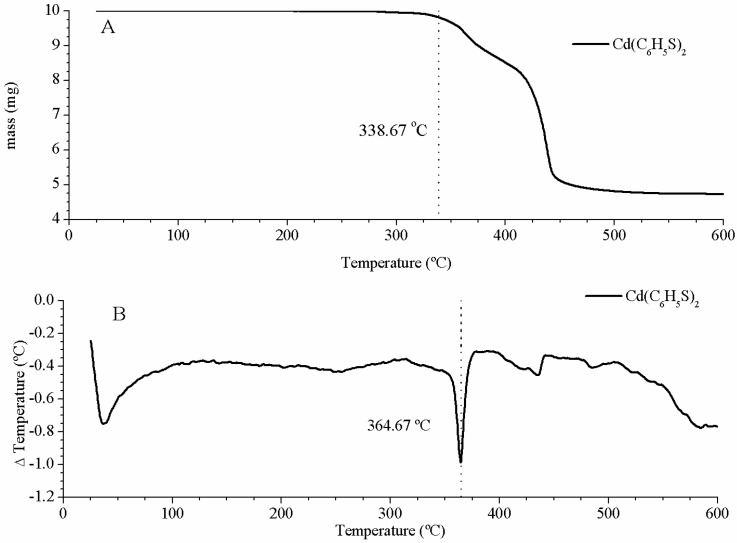
Cd(C_6_H_5_S)_2_—thermal gravimetric analysis (TGA) (**A**). Cd(C_6_H_5_S)_2_—differential scanning calorimetry (DSC) (**B**).

**Figure 6 nanomaterials-09-01212-f006:**
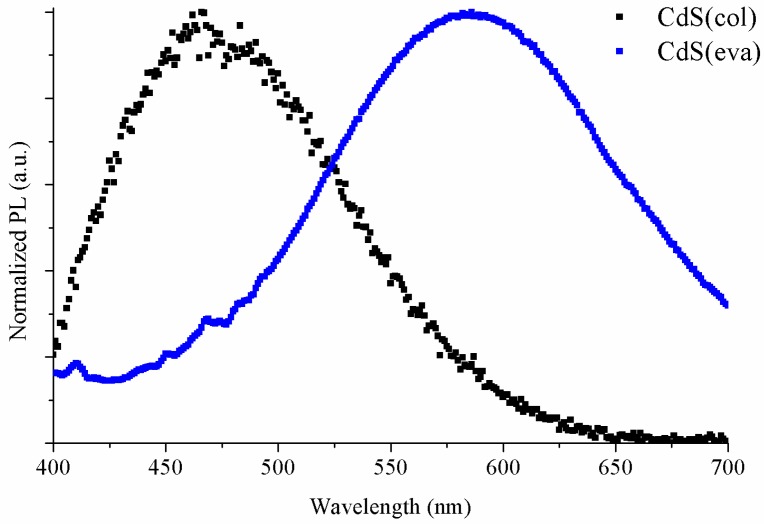
CdS(col) and CdS(eva) NCs normalized PL spectra.

**Figure 7 nanomaterials-09-01212-f007:**
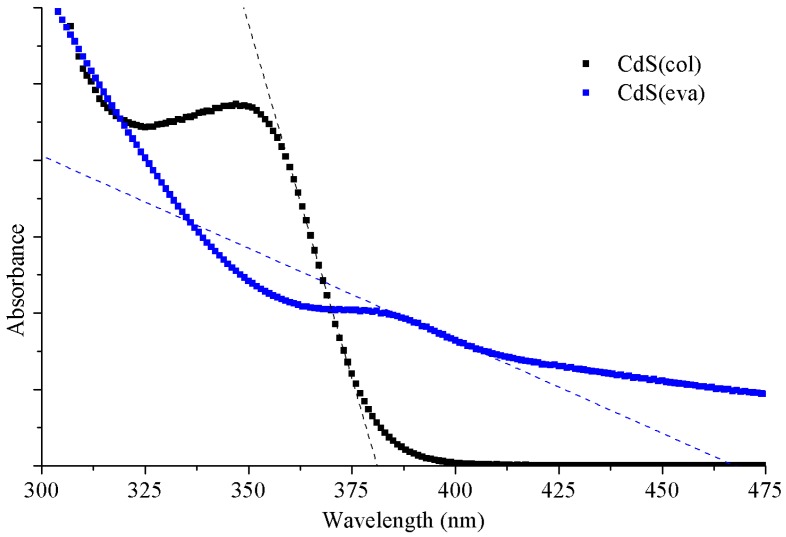
CdS(col) NCs (black) in DMSO and CdS(eva) NCs (blue) in toluene absorbance spectra.

**Figure 8 nanomaterials-09-01212-f008:**
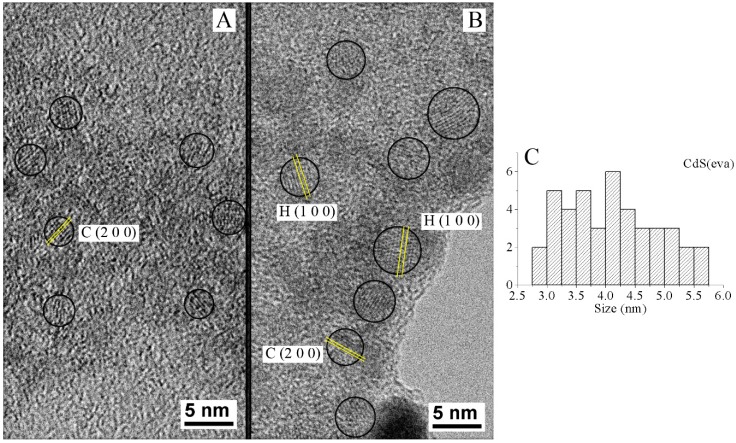
TEM images corresponding to (**A**) CdS(col) NCs and (**B**) CdS(eva) NCs. (**C**) Histogram for the calculation of the mean size and particle size distribution for CdS (eva) NCs.

**Figure 9 nanomaterials-09-01212-f009:**
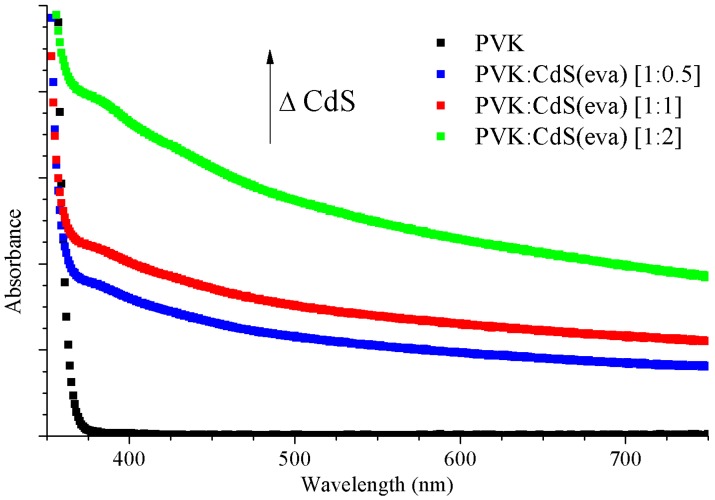
The absorption spectra of the hybrid solutions with increasing CdS content: PVK (black), PVK:CdS [1:0.5] (blue), PVK:CdS [1:1] (red) and PVK:CdS [1:2] (green).

**Figure 10 nanomaterials-09-01212-f010:**
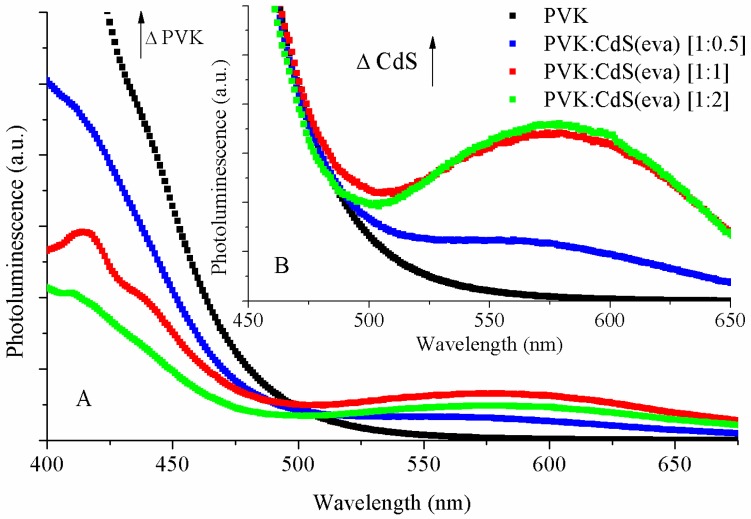
(**A**) The photoluminescence spectra and (**B**) the normalized PL spectra of the solutions: PVK (black), PVK:CdS(eva) [1:0.5] (blue), PVK:CdS(eva) [1:1] (red) and PVK:CdS(eva) [1:2] (green).

**Figure 11 nanomaterials-09-01212-f011:**
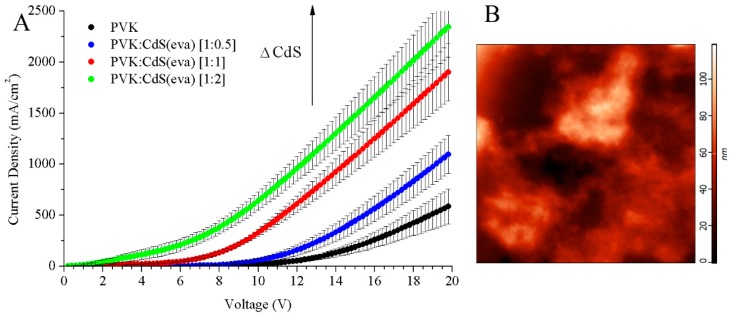
(**A**) J–V curves of the LEDs and (**B**) AFM image (area of 10 µm × 10 µm) corresponding to the hybrid LED with PVK:CdS(eva) [1:2] active layer. The color scale on the right indicates surface roughness.

**Figure 12 nanomaterials-09-01212-f012:**
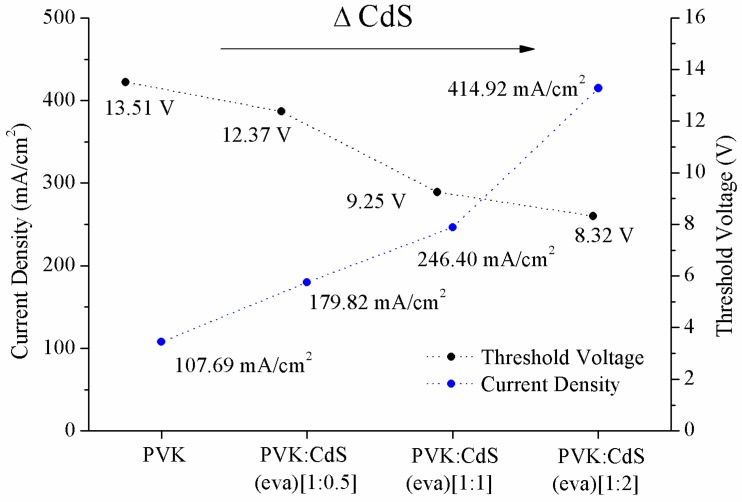
Threshold voltage (V) and Current density (mA/cm^2^) measured at threshold voltage for the LEDs from Figure 11A.

**Figure 13 nanomaterials-09-01212-f013:**
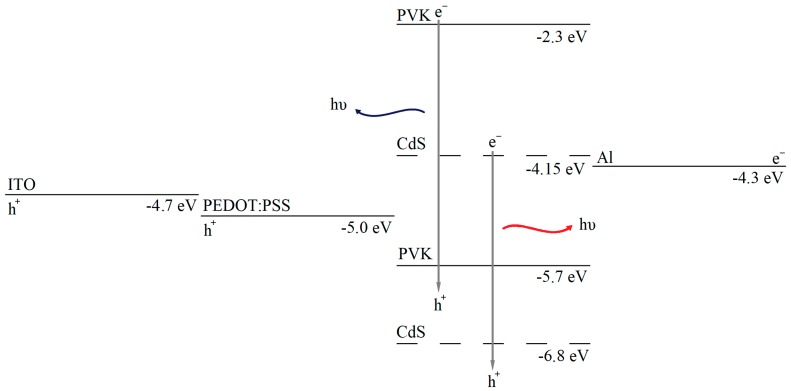
Energy diagram of ITO/PEDOT:PSS/PVK:CdS/Al device.

**Figure 14 nanomaterials-09-01212-f014:**
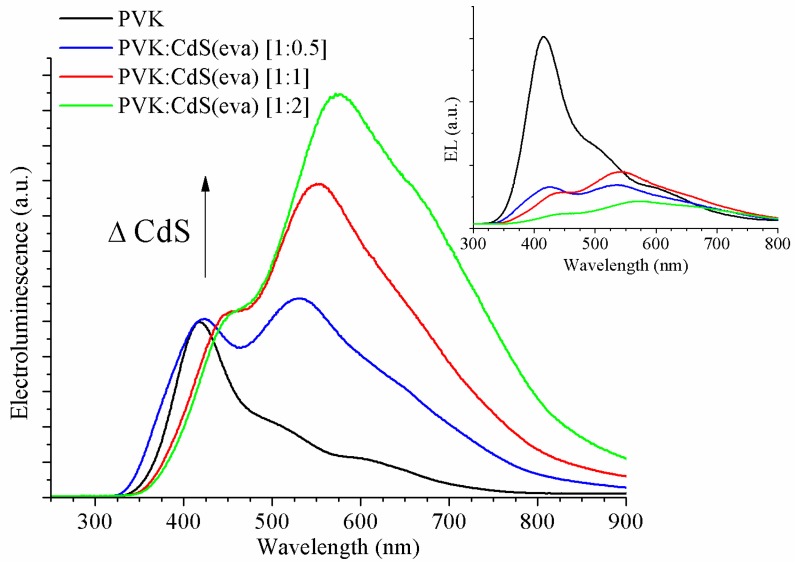
Normalized electroluminescence of the pristine PVK LED and hybrid LEDs using DMSO-free CdS NCs. PVK (black), PVK:CdS(eva) [1:0.5] (blue), [1:1] (red) and [1:2] (green). The inset shows as measured electroluminescence spectra for the same LEDs.

**Figure 15 nanomaterials-09-01212-f015:**
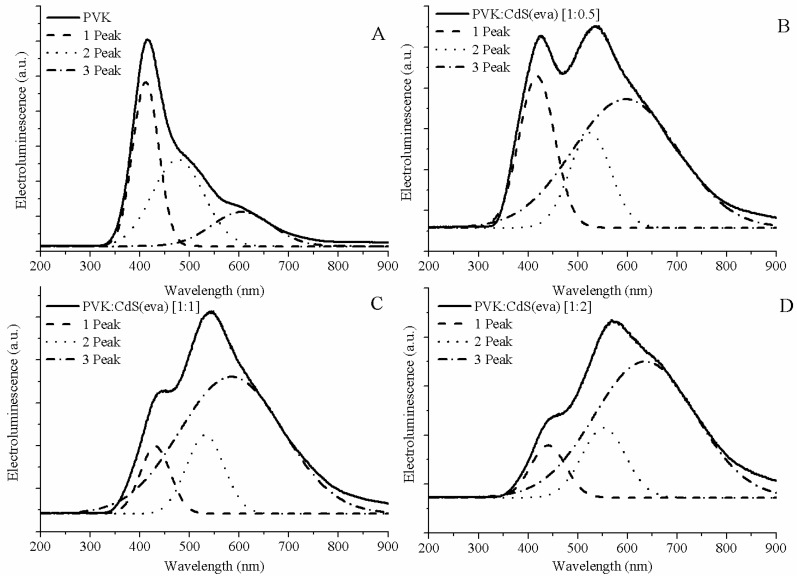
Gaussian deconvolution of electroluminescence spectra of PVK and hybrid LEDs. (**A**) Pristine PVK, (**B**) PVK:CdS(eva) [1:0.5], (**C**) PVK:CdS(eva). [1:1] and (**D**) PVK:CdS(eva) [1:2].

**Table 1 nanomaterials-09-01212-t001:** Comparison between colloidal CdS nanoparticles and evaporated CdS nanoparticles powder. Absorption edge values, wavelength of the maximum emission peak, diameter of the nanoparticles calculated by the Brus equation and measured from TEM images.

NCs	Absorption Edge		Emission Peak	Size (Brus)	Size (TEM)
CdS(col)	3.25 eV	381.2 nm	480 nm	2.9 nm	3.07 ± 0.07 nm
CdS(eva)	2.65 eV	467.6 nm	586 nm	4.9 nm	4.12 ± 0.12 nm

**Table 2 nanomaterials-09-01212-t002:** Stoichiometry of hybrid solutions prepared with evaporated CdS nanoparticles.

PVK:CdS(eva)	PVK	CdS(eva)
1:0	0.0401 g	—
1:0.5	0.0268 g	0.0134 g
1:1	0.0201 g	0.0201 g
1:2	0.0134 g	0.0268 g

**Table 3 nanomaterials-09-01212-t003:** Positions of the maximum of each emission peak for PVK–LED reference and the hybrid-LEDs doped with CdS nanoparticles. The wavelengths are obtained from the Gaussian deconvolution of electroluminescence spectra.

Hybrid-LEDs	Gaussian Emission Peaks		
PVK	412 nm	477 nm	606 nm
PVK:CdS(eva) [1:0.5]	418 nm	524 nm	596 nm
PVK:CdS(eva) [1:1]	431 nm	530 nm	605 nm
PVK:CdS(eva) [1:2]	441 nm	552 nm	636 nm

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
