# Peer review of "Expanded Electroluminescence in High Load CdS Nanocrystals PVK-Based LEDs"

_nanomaterials, 2019, doi:10.3390/nano9091212_

Round 1

Reviewer 1 Report

The authors have answered to all the referee comments, therefore I think that the paper can now be accepted.

Author Response

We are very grateful for your wise comments in order to improve our paper.

Reviewer 2 Report

The authors replied to the comments from the reviewers. I will add two small comments as follows.

1. Is the following expression common in the field?

m[Cd(C6H5-S)2]/V[DMSO] 90 = 40 mg/mL

2. The following expression is confusing. Which is which?

The efficiency of PVK and hybrid-LEDs doped with CdS NCs were 0.69 cd/A, 0.13 cd/A, 0.09 cd/A and 0.06 cd/A, respectively.

Author Response

Thank you very much for your careful revision. The answers to your questions are shown below:

To question 1.

Is the following expression common in the field? m[Cd(C6H5-S)2]/V[DMSO] 90 = 40 mg/mL

Although the expression pointed out for the reviewer might be not usual, we think it is very useful to easily report the ratio of solute mass to the solvent volume.

To question 2.

The following expression is confusing. Which is which? The efficiency of PVK and hybrid-LEDs doped with CdS NCs were 0.69 cd/A, 0.13 cd/A, 0.09 cd/A and 0.06 cd/A, respectively.

We have changed that sentence for a more detailed one as follows:

The efficiency of pristine PVK LED was 0.69 cd/A while for hybrid-LEDs doped with CdS NCs were 0.13 cd/A for [1:0.5], 0.09 cd/A for [1:1] and 0.06 cd/A for PVK:CdS(eva) [1:2].

Changes have been highlighted in the final version of the manuscript.

Reviewer 3 Report

The quality of the paper has been improved, in particular the comparison with literature data and the comments on the experimental sections. 

I only suggest some minor revisions:

in Figure 7, the label of y-axis has to be changed in "Absorbance" without "(a.u.)" since the absorbance is an a-dimensional quantity.  the authors should consider the paper "Chem. Mater. 2003, 15, 2854-2860" for the determination of the CdS size. For both samples, the sizes calculated with Brus formula and reported in Table 1 seem to be higher than those obtained using the formula proposed by Peng. In this case, the comparison with TEM data is less favorable. 

After minor revisions, I suggest the pubblication of the article on Nanomaterials journal. 

Author Response

Thank you very much for your suggestions.

The reviewer is right about the adimentionality of Absorbance. We have eliminated the (a.u.) legend in figures 1, 7 and 9.

Regarding the reference mentioned by the reviewer, we are aware of it. We have considered also its use for our nanocrystals, but the results obtained for evaporated nanocrystals were even  farther from TEM results than those obtained by using Brus method.

We have studied different models to calculate the size of nanocrystals, such as the sizing curves suggested by Peng (reference indicated by the reviewer), the hyperbolic band model, the Henglein formula or Brus equation. Brus was selected because his model fits better for a larger range of sizes. Besides, if the synthesis conditions are not the same, models based on empirical results could cause errors. Currently, we are working on a new paper where we compare different  methods to calculate the size of this type of nanocrystals and the ideal conditions to use each one of the commented models.

We are very grateful to the reviewer for helping to improve our manuscript.

This manuscript is a resubmission of an earlier submission. The following is a list of the peer review reports and author responses from that submission.

Round 1

Reviewer 1 Report

The authors report on a CdS Nanocrystals PVK based LEDs. They developed a method to make CdS nanoparticles capped thiophenol (dispersed in DMSO) compatible with PVK solution (in toluene), claiming that they could enhance LED morphology and electroluminescence.

I found that the paper has several major issues, hindering the publication on Nanomaterials.

Following some suggestion to improve the manuscript for a resubmission to a lower impact factor journal:

1) The developed procedure results in “powder” of NCs that is likely to form micrometer size aggregates, this explain the huge scattering readily observed in Fig 8. Are the dispersions of such a NCs stable? The authors should find a solution for this issue.

2) The authors should perform FT-IR to verify if the capping layer is removed or not after heating at 200°C.

3) The authors should explain why the emission in enhanced after heating.

4) To prove energy transfer the authors should measure PLE.

5) The major issue is the interpretation or results. I believe that the double peak in the EL spectra come from band edge emission of NCs plus a lower energy shoulder from intra-gap traps and not from cubic zinc-blend and hexagon wurzite structure, as the authors speculate. To prove that the lower energy EL band came from traps, I suggest measuring the EL emission as a function of voltage. Trap sites should get saturated at lower applied voltages, thus have a lesser influence on the emission observed at higher applied voltages.

6) I suggest the authors to evaluate the CIE coordinates of the device to check if it is white emitting, this can add some value to the work.

7) Manuscript is not well organized; some of the information should be moved to the “Materials and Methods section”.

8) Some minor mistake throughout the text, Organic light-emitting diodes (OLEDs) should be Hybrid light-emitting diodes.   

Reviewer 2 Report

This manuscript describes the preparation of CdS NCs (Nanocrystals, thiophenol capped) loaded in PVK layer, and their EL properties. Since NCs are dispersed DMSO, elimination of DMSO by heat treatment is effective to get good dispersion in the PVK layer. During this treatment, the size of the NCs are increased together with partial morphology change from cubic zinc-blende to hexagonal wurzite. EL spectral components were found to be divided into three elements. The author claimed that the shorter wavelength component is assigned to emission from cubic crystals, whereas the longer wavelength is assigned to be from hexagonal wurtzite.

Even though the authors shows many data, their discussion is seemingly not sufficiently supported by enough evidence. There are already three EL spectral components before the addition of CdS. Why are the components assigned indipenently to two emissions from two different crystals?

What is the efficiency of EL emission? Since the NCs are prepared in aqueous solution, the EL might be worse when compared with EL from NCs prepared in hydrophobic conditions. Are there any discussions on this issue?

Further, the manuscript has so many places to be improved.

In abstract, heating treatment à heat treatment

smaller cubic and coarser hexagonal -> smaller cubic and bigger hexagonal

In Fig. 1, the PL peak position is roughly 430 nm, not 480 nm.

In line 111, Table 1 is far away to find out.

In line 119, the following expression is quite awkward.

  Optical absorption and PL measurements of these solutions were taken.

In line 145, 3% by weight can be simply written as 3wt%.

In Fig.5, the indicator in the figure says that the one side is roughly 120 nm. But the caption says 10 micron.

In line 264, displacement should be shift.

In line 391, Gaussian decomposition should be Gaussian deconvolution.

Looking back to these issues, I am sorry to say, but I can not recommend the current manuscript to be published in Nanomaterials.

Reviewer 3 Report

The authors present a research on the inclusion of CdS nanoparticles on a PVK film for the preparation of a hybrid light emission device prototype. After reporting the characterization of the used materials, they analyzed the electronic and electroluminescence features of the final device. 

The manuscript can be improved considering also the following comments. 

1-    Authors should improve the introduction and the results sections with some comparison with other materials used for similar devices reported in literature

2-    The electroluminescence analysis could include also a luminescence efficiency value in order to have a better comparison with the performance measured with other hybrid LED

3-    The evaporation of DMSO and the further inclusion of CdS nanoparticles (NP) powder into the PVK solution could results in a heterogeneous dispersion of NP (also as aggregates) into the matrix. This can explain the high residual absorbance at high wavelengths, in particular for high NP concentration. Could this effect somehow alter the electroluminescence performances of the final system? Could the authors show some data obtained with NP having a better chemical affinity, through the use of appropriate capping agents, with the guest matrix? 

4-    On lines 363-364 the following sentence is written: “This ligands absence favours a direct connection between PVK and the CdS core, improving the electrical behaviour.” Could the authors be more specific? The meaning of this sentence is not clear.